# TabPrompt: Graph-based Pre-training and Prompting for Few-shot Table Understanding

**Rihui Jin**[1,2*]**, Jianan Wang**[3*]**, Wei Tan**[3]**, YongRui Chen**[1,2]**, Guilin Qi**[1,2]**, Wang Hao**[3†]

[1] School of Computer Science and Engineering, Southeast University, Nanjing, China
[2] Key Laboratory of New Generation Artificial Intelligence Technology and Its
Interdisciplinary Applications (Southeast University), Ministry of Education, China
[3] Alibaba Group
220212061@seu.edu.cn, {nanxi.wjn, celia.tw}@alibaba-inc.com,
{yrchen, gqi}@seu.edu.cn, wanghao.wh@alibaba-inc.com

## Abstract

Table Understanding (TU) is a crucial aspect of information extraction that enables machines to comprehend the semantics behind tabular data. However, existing methods of TU cannot deal with the scarcity of labeled tabular data. In addition, these methods primarily focus on the textual content within the table, disregarding the inherent topological information of the table. This can lead to a misunderstanding of the tabular semantics. In this paper, we propose TabPrompt, a new framework to tackle the above challenges. Prompt-based learning has gained popularity due to its exceptional performance in few-shot learning. Thus, we introduce prompt-based learning to handle few-shot TU. Furthermore, Graph Contrastive Learning (Graph CL) demonstrates remarkable capabilities in capturing topological information, making Graph Neural Networks an ideal method for encoding tables. Hence, we develop a novel Graph CL method tailored to tabular data. This method serves as the pretext task during the pre-training phase, allowing the generation of vector representations that incorporate the table's topological information. The experimental results of outperforming all strong baselines demonstrate the strength of our method in few-shot table understanding tasks.

## 1 Introduction

Abundant tabular data resources are readily available nowadays (Cafarella et al., 2008). However, the extensive knowledge within these corpora remains largely untapped because most tables are designed to be "human-friendly" rather than "machine-friendly" (Dong et al., 2019b). Consequently, to extract knowledge stored within these vast tables, it is crucial to enable machines to comprehend the semantics of tabular data, referred to as Table Understanding (TU). Furthermore, TU is fundamental in numerous subsequent tasks, including

Knowledge Base Augmentation (Bhagavatula et al., 2015) and Table QA (Zhang and Balog, 2020).

Early methods involved manual feature construction designed for specific table datasets (Chen and Cafarella, 2014) or utilizing CNN or LSTM architectures (Nishida et al., 2017). These methods demonstrate good performance only when there is an ample amount of manually labeled training data available. However, labeling tabular data requires high labor costs due to the inherent characteristics of tables, such as their two-dimensionality and flexible layouts. Consequently, these methods struggle to cope with the challenge posed by the **scarcity of labeled tabular data**.

More recently, researchers have explored the application of pre-trained language models to the task of TU (Wang et al., 2020b; Deng et al., 2020; Herzig et al., 2020). During the pre-training phase, the model learns optimal initialization parameters through self-supervised training on many unlabeled tabular data, enabling it to achieve excellent performance after fine-tuning downstream tasks. Although self-supervised pre-training reduces the burden of manual annotation to some extent, it still requires a substantial amount of labeled table data during the fine-tuning process. Thus, the existing "pre-training, fine-tuning" methods also remain insufficient in addressing the **scarcity of labeled tabular data** when performing TU tasks. In other words, existing methods struggle to perform well in few-shot TU.

On the other hand, tables are structured data that encompass not only textual content but also topological information related to their layout. However, the existing methods sometimes **misunderstand the tabular semantics resulting from disregarding the topological information within tables**. For example, many pre-training models (Herzig et al., 2020) flatten a table and concatenate its contents row by row into one-dimensional plain texts as the input of the encoder. These methods

---

[*] R. Jin and J. Wang contributed equally to this work.
[†] Corresponding author.

only focusing on textual content overlook the topological information of the table, as cells in the same column typically have semantic commonalities. As depicted in Fig. 1 (a), cells in the second column belong to the language category.

To address the above two challenges, we propose a new framework called TabPrompt. *First*, we incorporate soft prompts (Li and Liang, 2021) into the framework and devise a prompt-based learning method suitable for tabular data. Recently, the new paradigm of "prompt-based learning" has attracted extensive attention due to its remarkable performance in few-shot scenarios (Liu et al., 2021a). Prompt-based learning enables more effective knowledge transfer by making downstream TU tasks more compatible with the pre-training task. By leveraging it, TabPrompt is capable of effectively tackling the challenges posed by few-shot TU. *Second*, we employ a novel Graph Contrastive Learning (Graph CL) to encode tabular data during pre-training. GNNs are well-suited for processing data with topology (Kipf and Welling, 2016), making them an ideal choice for capturing the topological information of tabular data. In addition, to encode the intrinsic structure knowledge, CL is one the most effective and popular pretext tasks (Zhang et al., 2020). In this regard, TabPrompt introduces a graph constructing method that fully considers the internal relationship between cells, which enables TabPrompt to better learn the topological structure information of the table during pre-training. To evaluate TabPrompt, we conduct extensive experiments in few-shot scenarios on three public datasets by comparing TabPrompt against several strong baselines. The results of outperforming all the baselines demonstrate the effectiveness of TabPrompt in the few-shot scenarios.

The contributions of this paper are summarized in the following points:

- We apply prompt-based learning to the TU task to tackle the scarcity of labeled tabular data. To the best of our knowledge, this is the first attempt to introduce prompt learning into the field of TU.
- To obtain vector representations that incorporate the topological information of tables, we introduce a novel Graph CL method as the pretext task to pre-train the encoder GNN.
- To evaluate TabPrompt, we conduct experiments on publicly available datasets, focusing on two specific few-shot TU sub-tasks. In

both tasks, the results of TabPrompt outperforming all baselines underscore its superiority in handling few-shot TU scenarios.

## 2 RELATED WORKS

**Graph Neural Network (GNN).** GNN architectures, such as GCN (Kipf and Welling, 2016) and GIN (Xu et al., 2018), have gathered substantial interest among researchers for their remarkable capability to handle real-world data containing inherent topological structure. Recognizing that tables inherently embody topological information, several studies (Du et al., 2021; Wang et al., 2021) have explored the application of GNNs in table-related research fields. However, their methods of constructing the tabular graph fail to fully consider the topological relationships between different cell types, resulting in the misunderstanding of tabular semantics.

**Pre-training and Fine-tuning.** Since the introduction of BERT, the "pre-training and fine-tuning" paradigm has gained significant attention. Researchers have adapted this paradigm to TU by designing customized pretext tasks tailored to tabular data (Wang et al., 2020b; Iida et al., 2021; Deng et al., 2020). While these methods effectively leverage unlabeled data during pre-training, they still rely on a substantial amount of labeled data in the fine-tuning stage to achieve optimal performance. As a result, these methods struggle to handle few-shot TU scenarios where only a limited amount of labeled data is available. Additionally, these methods often flatten the table into a sequential input, disregarding the inherent topology of the table. This can lead to the loss of crucial topological information during the modeling process.

**Other Deep Learning-based TU.** In the early stages of table understanding (TU) research, the emphasis was primarily on Cell Entity Linking (Ibrahim et al., 2016; Hassanzadeh et al., 2015; Efthymiou et al., 2017; Bhagavatula et al., 2015). These methods often relied on pre-defined ontologies and external knowledge bases, which limited their versatility. However, more recent work has shifted towards addressing broader TU tasks such as Table Cell Classification (Ghasemi-Gol et al., 2019; Sun et al., 2021) and Table Type Classification (Eberius et al., 2015; Nishida et al., 2017). These methods typically utilize manual features or early neural architectures like LSTM, which need a large quantity of manually labeled data for training.

List of most commonly learned foreign languages in the U.S.

| rank | language | total enrollments | percentage |
|---|---|---|---|
| 1 | Spanish | 822,985 | 52.20% |
| 2 | French | 206,426 | 13.10% |
| 3 | German | 94,264 | 6% |
| 4 | American Sign | 78,829 | 5% |
| 5 | Italian | 78,368 | 5% |
| 6 | Japanese | 66,605 | 4.20% |
| 7 | Chinese | 51,582 | 3.30% |
| 8 | Arabic | 23,974 | 2% |

(a) A sample table in TURL

Net lending/borrowing & primary balance in Italy

| Years | Net lending/ borrowing | Primary balance |
|---|---|---|
| 1995 | -7.3 | 3.8 |
| 1996 | -6.6 | 4.5 |
| 1997 | -3.0 | 6.2 |
| 1998 | -3.0 | 4.9 |
| 1999 | -1.8 | 4.6 |
| 2000 | -1.3 | 4.8 |
| 2001 | -3.4 | 2.7 |
| 2002 | -3.1 | 2.4 |

| index |
|---|
| indexName |
| value |
| valueName |

(b) A sample table in WebSheet

Figure 1: A subtask of TU: Cell Type Classification. Cells of different colors represent different cell types.

| ISBN | 312424094 |
|---|---|
| ISBN-13 | 9780312424091 |
| Pages | 224 |
| Publisher | Picador USA |
| Published | 2004 |
| Language | English |
| Alibris ID | 12873728485 |

(a) entity table

| Authors Name |
|---|
| Raadik, Tarmo |
| Raadik, Tarmo A. |
| Raas-Rothschild, A |
| Raasch, Beverly |
| Rabaa'i, Ahmad A. |
| Rabago, D. |

(b) list table

| Size | Waist |
|---|---|
| Small | 28" - 30" |
| Medium | 30" - 32" |
| Large | 32" - 35" |
| XLarge | 36" - 39" |
| XXLarge | 40" - 44" |

(c) relational table

| Buy PbNation T-Shirts |
|---|
| About Us |
| FAQ |
| Staff |
| Members |
| Leaderboard |

(d) non-data table

Figure 2: A subtask of TU: Table Type Classification. Examples of table types from WCC.

**Prompt-based Learning.** The main idea of the new paradigm "pre-train, prompt, and predict" (Liu et al., 2021a) is to reformalize the target task to look more like the pretext task to use better what the model has already learned. An increasing number of novel prompt-based learning methods have been proposed, encompassing various forms of prompts, such as soft prompts (Li and Liang, 2021). However, it is worth noting that there is currently no research available on the development of prompts specifically designed for tabular data.

## 3 PROPOSED METHOD

In this section, we first introduce the preliminaries of TU sub-tasks relevant to our work and then describe TabPrompt in detail. In this section, we begin by introducing the preliminaries of TU sub-tasks relevant to our work. Subsequently, we present a comprehensive description of TabPrompt.

### 3.1 Preliminaries

Given a table $T = \{c_{i,j}|0 \leq i < N, 0 \leq j < M\}$ where $N$ is the number of rows, $M$ is the number of columns, and $c_{i,j}$ is the cell located in the $i_{th}$ row and $j_{th}$ column.

**Cell Type Classification (CTC).** This sub-task of TU involves the identification of the cell type for each cell $c_{i,j}$ in a table $T$. CTC has been widely studied by many works in which different taxonomies of cell types were used (Wang et al.,

2020b; Du et al., 2021; Sun et al., 2021). In our work, we adopt the taxonomy used in Dong et al. 2019a and expand it by including an additional type for a comprehensive comparison, as shown in Fig. 1. Definitions of four cell types are as follows: A **value** represents a basic unit describing the content within a table. A **valueName** serves as a summary unit of **value** cells in the same column. An **index** is utilized to index **value** cells. An **indexName** acts as a summary unit for **index** cells in the same column.

**Table Type Classification (TTC).** This task involves table-level categorization, requiring machines to classify tables based on certain taxonomy. Similar to CTC, there are multiple taxonomies in this task. We follow the taxonomy introduced in Wang et al. 2020b as illustrated in Fig. 2. The taxonomy consists of four types: **relational**, **entity**, **list**, and **non-data**.

### 3.2 An Overview of TabPrompt

An overview of our framework is shown in Fig. 3 First, TabPrompt transforms the tabular data into graph data, taking into account the topological relationships between cells during the graph construction. Second, TabPrompt utilizes the tabular Graph CL as the pretext task to pre-train the encoder GNN. Lastly, TabPrompt is trained on a limited amount of labeled data to tune soft prompts. This is accomplished by reformulating the objectives of CTC and

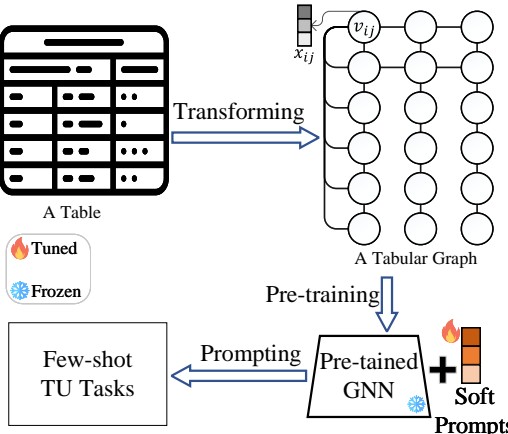

Figure 3: An Overview of TabPrompt.

TTC, aiming to bridge the gap with the pre-trained objective.

## 3.3 Construct the Tabular Graph

Existing methods that utilize GNNs often consider adjacent table cells or cells with similar strings as neighboring nodes in the graph (Du et al., 2021; Wang et al., 2021). However, these methods fail to adequately capture the internal topological relationships between nodes of different types. In order to address this limitation, we adopt a more sophisticated method of constructing graph data. We establish connections between cells that exhibit specific topological relationships. This transformation method aligns with the fundamental assumption of our tabular Graph CL method, which posits that connected nodes should have closer relationships and display higher similarity in their vector representations.

According to our observations of tabular data, we identify several characteristics of table cells: 1) Cell pairs within the same column exhibit higher levels of relatedness compared to cell pairs across different columns. 2) Cells within the header row of a table tend to have similar levels of dependencies. 3) Each individual cell within a merged cell is the same type. 4) The string within the header cell often serves as a summary description for the non-header cells in the same column. Based on these patterns, we establish links between cells in the following situations: 1) Adjacent cells within the same column. 2) Adjacent cells within the header row. 3) Cells split from merged cells. 4) Cells within the header row and non-header cells within the same column.

## 3.4 Tabular Graph Pre-training

In this section, we initially introduce our tabular Graph CL employed for pre-training, followed by a detailed discussion of the pre-training process.

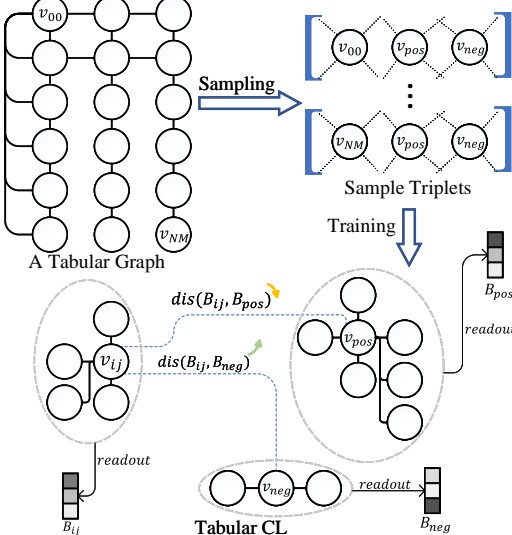

Figure 4: Tabular Graph Contrastive Learning.

**Tabular Graph Contrastive Learning.** After tables are transformed into graphs, our graph CL tailored to tabular data can be carried out. The process of our tabular graph CL method is shown in Fig. 4.

A tabular graph of table $T$ can be defined as $T_g = (V, E)$, where $V$ is the set of nodes representing cells $C$ in table $T$. Let $x_{ij} \in \mathbb{R}^d$ denote the embedding of node $v_{ij} \in V$. $x_{ij}$ consists of the output of the string in the cell $c_{ij}$ through BERT (Devlin et al., 2019) and some handcrafted features that are general on tabular data. We observe that the information carried by surrounding cells may be helpful in determining the identity of the central cell. For example, if a cell is surrounded by numeric cells, the cell is also likely to be numeric. We define $B_{ij}$ as the embedding of cell $c_{ij}$ combining with neighbors' information. The formula of $B_{ij}$ is as follows:

$$B_{ij} = readout(\{x_{ij}, X_{N(x_{ij})}\}), \qquad (1)$$

where $N(x_{ij})$ is the cell set containing adjacent cells of $x_{ij}$. $X_{N(x_{ij})}$ is the embedding set of cells in $N(x_{ij})$. Note that $B_g$ is the vector representation of a whole tabular graph $T_g$. The choice of $readout$ function is flexible such as sum pooling, mean pooling, or concatenation.

Our tabular Graph CL method follows the assumption that the distance of connected cell pairs

should be closer than that of unconnected cell pairs in the embedding space. Formally, given a sample triplet of cells $(c_{ab}, c_{pos}, c_{neg})$ that $(c_{ab}, c_{pos}) \in E$ and $(c_{ab}, c_{neg}) \notin E$, the vector distance between the former cell pair should be smaller than that of the latter as follows:

$$dis(B_{ab}, B_{pos}) < dis(B_{ab}, B_{neg}), \quad (2)$$

where $dis$ is the reciprocal of the cosine distance function.

**Pre-Training.** The strength of pre-trained models lies in their ability to leverage a vast amount of unlabeled data. Similarly, in our tabular Graph CL, we capitalize on its label-free nature by employing it as the pretext task for model pre-training. During this phase, the model learns to assign similar vector representations to two highly related nodes. After pre-training, we can determine whether there is a high degree of correlation between two cells by measuring the vector distance between them.

Before initiating the pre-training process of the model, it is necessary to prepare the training data. The training data comprises triples that consist of nodes along with their corresponding positive and negative nodes. Formally, given a tabular graph $T_g$, we sample a pair of positive and negative cells for every cell in $T_g$ to form a triplet $(c_{ij}, c_{pos}, c_{neg})$ for CL. To construct a sample set $\mathcal{S}$ for pre-training, we sample triplets from every graph $T_g$ in the tabular graph set $\mathcal{T}_G$. We employ GNN as the pre-trained encoder with its parameters $W_i$. The objective function of pre-training is as follows:

$$Z(ij, pos, neg) = exp(dis(B_{ij}, B_{pos})/\tau) \quad (3)$$
$$+ exp(dis(B_{ij}, B_{neg})/\tau),$$

$$\mathcal{L}_{pre}(W_i) = \sum_{\substack{(c_{ij}, c_{pos}, c_{neg}) \in \mathcal{S} \\ (c_{ij}, c_{pos}) \in E \\ (c_{ij}, c_{neg}) \notin E}} \ln \frac{exp(dis(B_{ij}, B_{pos})/\tau)}{Z(ij, pos, neg)}, \quad (4)$$

where $\tau$ is a temperature hyperparameter often used in CL to adjust the distribution shape.

When pre-training ends, the encoder GNN is utilized to handle downstream TU tasks with the pre-trained parameters $W_i$.

### 3.5 Tabular Graph Prompting

Prompting-based learning aims to bridge the significant gap between pre-training and downstream tasks, which allows the model to transfer the knowledge acquired during pre-training more effectively.

In this section, we introduce the process of reformulating two TU tasks to enhance their compatibility with the pre-training objective, as well as the details for tuning the soft prompts. Fig. 5 shows the process of prompting for CTC.

**Prompt Addition.** In the prompting stage, a key step is to reformulate the raw input by designing the appropriate prompt addition (Liu et al., 2021a). For instance, in machine translation, the input "I love you." is reformulated as "English: I love you. French: [Z]" with [Z] denoting an answer slot. Nevertheless, applying such templates to tabular data is challenging due to the interdependence of text within adjacent cells. Therefore, we develop a customized prompt addition specifically designed for tabular data.

As mentioned above, our pre-trained model is inclined to give closer vector representations to highly related cells when presented with a table as input. Consequently, the machine can determine whether two nodes share the same type based on the distance between their embeddings. Following this idea, we create proxy cells for each cell type, with their vector representations calculated as the mean of the vector representations of the same type of nodes in the few-shot training set. For each cell in a table, the machine compares its vector representation with those of all proxy nodes to determine its type. Similar prompt addition is also employed for the TTC task, the added proxy node represents the type of the table, and its vector is calculated as the mean of the $B_{T_g}$ of tables of the same type.

**Prompt tuning.** In the prompt-based training process, there is a potential risk of overfitting in few-shot experimental settings if all parameters are tuned (Dong et al., 2021). To mitigate this, we freeze the pre-trained parameters $W_i$ and introduce the soft prompt (Fang et al., 2022). This involves employing the learnable prompts $\mathbf{p}_c$ and $\mathbf{p}_t$ for the CTC and TTC tasks, respectively. We perform the element-wise multiplication between the vector of each node and the corresponding soft prompt. This operation can be seen as assigning weights to each element of the vector. By tuning the learnable soft prompts, the model can assign larger weights to elements that are more relevant to the specific tasks, thereby enhancing its performance.

Formally, given the CTC dataset $D_t^C$ containing a labeled set of tabular graphs $T_g'$ where each cell $c_{ij}$ is associated with a label $y_{c_{ij}}^C$, we denote the label set for the CTC task as $Y^C = \{y_0^C, ..., y_4^C\}$.

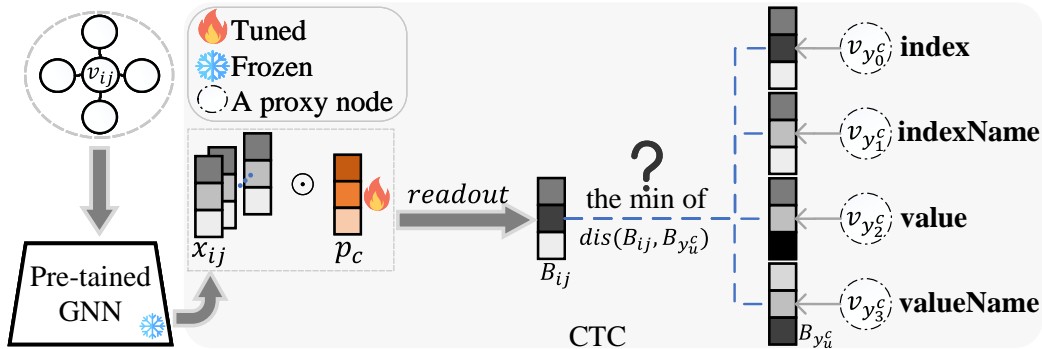

Figure 5: Prompting for CTC.

The formulas for training $\mathbf{p}_c$ are as follows:

$$B_{ij}^C = readout(\{\mathbf{p}_c \odot x_{uv} | x_{uv} \in N_{x_{(ij)}}\}), \quad (5)$$

$$\mathcal{L}_{pro}(\mathbf{p}_c) = \sum_{\substack{T_g \in \mathcal{D}_t^C \\ (c_{ij}, y_{c_{ij}}^C) \in T_G'}} \ln \frac{exp(dis(B_{ij}^C, B_{y_{c_{ij}}^C})/\tau)}{\sum_{u=0}^{4} exp(dis(B_{ij}^C, B_{y_u^C})/\tau)}, \quad (6)$$

where $B_{y_u^C}$ is the vector of the proxy node representing the table cell labeled as $y_u^C$. Similarly, the TTC dataset $D_t^T$ contains a set of labeled tabular graph $T_g' = (V, E, y_g^T)$ where $y_g^T$ is the label of its table type. The label set of our TTC task is denoted as $Y^T = \{y_0^T, ..., y_4^T\}$. The formulas for training $\mathbf{p}_t$ are as follows:

$$B_g^T = readout(\{\mathbf{p}_t \odot x_{uv} | x_{uv} \in V\}), \quad (7)$$

$$\mathcal{L}_{pro}(\mathbf{p}_t) = \sum_{(T_g', y_g^T) \in \mathcal{D}_t^T} \ln \frac{exp(dis(B_g^T, B_{y_g^T})/\tau)}{\sum_{u=0}^{4} exp(dis(B_g^T, B_{y_u^T})/\tau)}, \quad (8)$$

where $B_{y_u^T}$ is the vector of the proxy node representing the table labeled as $y_u^T$.

## 4 EXPERIMENTS

In this section, we conduct experiments on public datasets to evaluate the performance of TabPrompt in both the CTC and TTC tasks.

### 4.1 Datasets

Although our method can be applied to tables from other documents (CSV sheets, PDF documents, etc.), we focus on web tabular data in this paper since they are easily accessible and easy to parse. We employ four datasets in our work. 1) *TabEL* (Bhagavatula et al., 2015) contains around 1.6M tables extracted from Wikipedia pages. 2) *TURL* (Deng et al., 2020) selected 670k relational tables

from *TabEL* and annotated the column number corresponding to the subject column of each relational table. 3) *WebSheet* (Dong et al., 2019a) contains about 3k tables with cell type labels which were selected and manually annotated from web-crawled Websheet (only 50 of them publicly available). 4) *WCC* (Ghasemi-Gol and Szekely, 2018) is a sample of *July 2015 Common Crawl* and is annotated with the web table taxonomy introduced by Crestan and Pantel 2011. The taxonomies adopted in these datasets differ slightly from the taxonomy used in this paper. We comprehensively describe the correspondence between the various taxonomies and other details in the appendix A.

### 4.2 Baselines

We compare TabPrompt with strong baselines for CTC and TTC to verify the effectiveness of TabPrompt.

We employ the following baselines for CTC. TCC-Embd (Ghasemi-Gol et al., 2019) is a tabular cell classification method with pre-trained CBOW&Skip-gram cell embeddings (Mikolov et al., 2013). PSL (Sun et al., 2021) reformalizes the CTC task into block detection, which aims to detect the data blocks in the table. TabularNet (Du et al., 2021) utilizes a homogeneous graph constructed out of the WordNet (Fellbaum, 2000) knowledge base and adopts GCN as the encoder and LSTM. TableFormer (Yang et al., 2022) is guaranteed to be invariant to row and column order perturbations. Because it is specifically for Table QA and Table Verification, it means that it cannot handle the TU directly. To adapt it for TU, we treat these models as encoders that output embeddings of table cells and stack layers of downstream classifiers (including an MLP layer and a pooling layer) on top of the encoders. In that case, these methods can be compared with our method. FewTPT (Liu et al., 2022) is a Chinese tabular language

model, thus, we pre-train it using 570k Wiki tables in English from scratch. While the "Table Classification" mentioned in FewTPT centres on identifying header domains within tables, distinct from the focus of TTC, we employ a similar processing methods as in the case of TableFormer to facilitate handling the TU addressed in this paper. TUTA (Wang et al., 2020b) is a pre-training model with tree-based transformers for TU, which achieves SOTA performance among existing methods.

We employ the following baselines for TTC. DWTC (Eberius et al., 2015) train a Random Forest model through manually engineered features. TabNet (Nishida et al., 2017) utilizes a hybrid deep neural network architecture of LSTM and CNN. TabVec (Ghasemi-Gol and Szekely, 2018) is an unsupervised method to embed tables based on table-level manual features. Additionally, when considering TTC, TableFormer and FewTPT can also be assessed by applying stacked downstream classifiers. In addition to CTC, TUTA (Wang et al., 2020b) also performs well on TTC.

### 4.3 Settings and parameters

The embedding of a node in a tabular graph is composed of semantic information and manual features. The embedding of semantic information is obtained by feeding the string of the node into Sentence-BERT (Reimers and Gurevych, 2019) and then performing dimensionality reduction. The manual features used in our work are introduced in Crestan and Pantel 2011, which have strong versatility on tabular data. We present details of the manual features in Appendix B. To be fair, the manual features used in TabPrompt are also used by all baselines. We employ the macro-F1 score and the standard deviation as the evaluation metric in our experiments, commonly used in other TU works.

In the pre-training phase, we employ a 3-layer GIN as the backbone and set the hidden dimension as 64. We randomly sample 5k tables from *TabEL* and utilize the tabular Graph CL method mentioned above as the pretext task to pre-train the parameters of the backbone. The details of hyper-parameters can be found in Appendix C.

Following a typical k-shot classification setting (Liu et al., 2021b; Zhou et al., 2019; Wang et al., 2020a), we generate a series of few-shot downstream tasks of CTC and TTC for model training, validation, and testing.

For CTC, we conduct this downstream task on

Table 1: Macro-F1 evaluation on CTC.

| Methods | TURL | | WebSheet | |
|---|---|---|---|---|
| | 1-shot | 3-shot | 1-shot | 3-shot |
| TCC-Embd | $26.72 \pm 6.27$ | $31.14 \pm 8.51$ | $19.62 \pm 12.54$ | $21.42 \pm 12.39$ |
| PSL | $29.23 \pm 7.55$ | $39.72 \pm 7.43$ | $15.53 \pm 10.11$ | $18.55 \pm 13.18$ |
| TabularNet | $34.41 \pm 7.47$ | $45.14 \pm 4.77$ | $20.64 \pm 12.92$ | $29.44 \pm 12.56$ |
| TableFormer | $33.68 \pm 6.33$ | $45.78 \pm 5.88$ | $21.74 \pm 8.95$ | $27.75 \pm 7.15$ |
| FewTPT | $33.21 \pm 10.22$ | $44.71 \pm 8.13$ | $22.93 \pm 9.75$ | $28.41 \pm 8.42$ |
| TUTA | $35.73 \pm 6.81$ | $47.32 \pm 9.48$ | $27.65 \pm 13.25$ | $34.77 \pm 9.84$ |
| TabPrompt | $\mathbf{46.82} \pm 8.80$ | $\mathbf{57.79} \pm 9.01$ | $\mathbf{30.78} \pm 10.25$ | $\mathbf{35.96} \pm 9.76$ |

two datasets, i.e., *TURL* and *WebSheet*. On each dataset, we generate ten $z$-shot Cell Type Classification tasks for training and validation. In each task, $z$ tables are randomly sampled for both training and validation, where $z \in \{1, 3\}$. Additionally, we randomly generate 20 tables for testing from the remaining not sampled for training and validation. After conducting the experiments, we calculate the mean macro-F1 score of the ten testing results, along with the standard deviation. For TTC, following a similar process of CTC, we randomly generate 10 $z$-shot ($z \in \{3, 5\}$) table classification tasks (i.e., sample $z$ tables per class in one task) from *WCC* and the remaining tables not sampled for testing.

### 4.4 Performance Evaluation

In this section, we analyze the results of experiments of CTC and TTC conducted on public datasets, respectively.

**CTC.** Based on the results of the few-shot CTC presented in Table 1, we draw the following conclusions. Firstly, the superior performance of TabPrompt over all baselines underscores the effectiveness of our proposed framework. This result proves that prompt-based learning is well-suited for addressing few-shot TU scenarios. Secondly, despite TUTA having a higher number of parameters pre-trained with more data, TabPrompt achieves better performance. This finding highlights the significance of bridging the gap between pre-training and downstream tasks, as it enhances the effectiveness of knowledge transfer from pre-training to downstream tasks. Thirdly, TabPrompt outperforms TabularNet, indicating that considering the topological information within the table significantly aids the model in better comprehending the table's semantics. Fourthly, TableFormer and FewTPT exhibit inferior performance compared to TUTA despite all being pre-trained. This is mainly because neither are designed for the TU task, leading to the absence of modules tailored to TU tasks.

Table 2: Macro-F1 evaluation on TTC.

| Methods | WCC | |
| --- | --- | --- |
| | 3-shot | 5-shot |
| DWTC | $26.12 \pm 9.71$ | $31.52 \pm 10.01$ |
| TabNet | $14.53 \pm 13.76$ | $16.35 \pm 10.17$ |
| TabVec | $31.43 \pm 8.63$ | $49.12 \pm 10.26$ |
| TableFormer | $42.14 \pm 6.73$ | $50.17 \pm 6.18$ |
| FewTPT | $44.97 \pm 7.13$ | $52.72 \pm 6.42$ |
| TUTA | $46.12 \pm 8.73$ | $54.45 \pm 8.27$ |
| TabPrompt | $\mathbf{50.75} \pm 7.18$ | $\mathbf{57.41} \pm 5.14$ |

Table 3: Ablation results.

| | CTC (TUTA) 3-shot | CTC (WebSheet) 3-shot | TTC (WCC) 5-shot |
| --- | --- | --- | --- |
| TabPrompt | $\mathbf{57.79} \pm 9.01$ | $\mathbf{35.96} \pm 9.76$ | $\mathbf{57.41} \pm 5.14$ |
| $w/o$ cl | $46.12 \pm 10.48$ | $31.52 \pm 9.01$ | $54.52 \pm 8.76$ |
| $w/o$ sp | $55.74 \pm 11.16$ | $33.25 \pm 9.87$ | $56.14 \pm 9.11$ |
| $w/o$ pr | $49.15 \pm 9.38$ | $30.18 \pm 10.47$ | $52.32 \pm 13.5$ |

**TTC.** We present the results of few-shot TTC in Table 2. First, the consistently superior performance of TabPrompt once again demonstrates the effectiveness of our proposed framework. Second, as both CTC and TTC share the same parameters of the pre-trained model, the superior performance of GraphPrompt on both types of tasks further supports the notion that our tabular Graph CL enables the models to learn better vector representations.

**Performance with different shots.** From Table 1 and Table 2, it is evident that the performance of all models improves as the number of samples increases. Therefore, we study the trends in performance with varied shots. We have Method A, Method B, and our method tested with different shots on the *TURL* dataset in Fig 6. The graph shows that in the case of low shots, TabPrompt consistently outperforms other methods. Although TabPrompt is surpassed when it comes to 50 shots (typically beyond 600 table cells), the amount of data is beyond the scope of our target scenario.

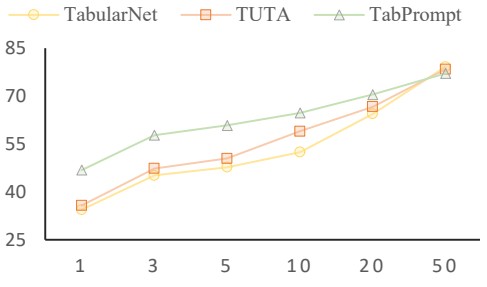

Figure 6: Impact of varied shots on CTC.

### 4.5 Ablation study

To analyze the individual contribution of each component in TabPrompt, we conduct the following ablation study:

- $w/o$ cl: When constructing graph data without taking into account the topological information of the table, link all adjacent cells simply.
- $w/o$ sp: Downstream tasks are performed without tuning the parameters of the soft prompt.
- $w/o$ pr: Using fine-tuning instead of prompt tuning to train the model in downstream tasks.

Results of ablation experiments are shown in Table 3. It is evident that TabPrompt's performance significantly deteriorates when using a vanilla method to construct graph data. This can be attributed to the fact that the vanilla method dramatically diminishes the effectiveness of tabular Graph CL. It confirms that considering the topological information within tables aids machines in comprehending the semantics of tables more effectively.

In our previous analysis, we hypothesize that the soft prompt can assign more weight to distinguish features during the prompting process. We observe that when the parameters of the soft prompt are not tuned, the performance of TabPrompt experiences a slight decrease. This observation confirms our hypothesis.

When we replace the prompt tuning method with fine-tuning, we observe a decrease in the model's performance across all three datasets. This outcome further reinforces that prompt tuning can effectively enhance the model's performance in scenarios with limited training tabular data.

### 4.6 Case Study

To facilitate an intuitive interpretation, we present a typical case in the CTC testing data that demonstrates the functionality of tabular Graph CL.

Fig. 7 shows two classification results for a table. Fig. 7 (a) represents a correct CTC result, whereas the classification results in Fig. 7 (b) have some flaws. It highlights that misunderstanding may result from neglecting the topological structure information within the table, such as the cells in one column tending to be a category as well as that cells in the header rows being more likely

to be a **valueName** rather than a **value**. For instance, the misclassified cell "July 1780 Typhoon" should have a stronger relationship with the upper and lower cells, while the relationship with the right cell "1780" is relatively weak despite both containing the textual content of "1780".

| indexName | index | valueName | value |
|---|---|---|---|
| Deadliest Pacific typhoons | | | |

| typhoon | year | fatalities |
|---|---|---|
| " Haiphong " | 1881 | 300,000 |
| Nina | 1975 | 229,000 |
| July 1780 Typhoon | 1780 | 100,000 |
| " Swatow " | 1922 | 60,000 |
| "China" | 1912 | 50,000 |

(a) CTC results of TabPrompt

| typhoon | year | fatalities |
|---|---|---|
| " Haiphong " | 1881 | 300,000 |
| Nina | 1975 | 229,000 |
| July 1780 Typhoon | 1780 | 100,000 |
| " Swatow " | 1922 | 60,000 |
| "China" | 1912 | 50,000 |

(b) CTC results of TabPrompt $w/o$ cl

Figure 7: A real case for CTC.

## 5  CONCLUSION

In this paper, we proposed a new framework TabPrompt. To tackle the scarcity of labeled tabular data and capture the untapped topological information, we resorted to prompt-based learning and Graph CL, respectively. The experimental results of outperforming all baselines demonstrate the effectiveness of TabPrompt in few-shot TU.

In future work, we aim to extend our method to handle domain-specific tables and more complex table structures, further enhancing its capabilities. In addition, we introduce a multimodal technique that leverages both text and image data within the table to enhance the understanding of its semantics.

## Limitations

First, the input of TabPrompt must be a machine-parsable data format, such as HTML files with <table> tags or CSV files. TabPrompt does not encompass the processes of locating or extracting tables from images or original web pages. Second, TabPrompt cannot handle tables with overly complex layouts, such as tables with images or too many empty cells. Third, TabPrompt performs poorly on small-sized tables due to the limited amount of text information available within these tables.

## Acknowledgement

This work is partially supported by National Nature Science Foundation of China under No. U21A20488. We thank the Big Data Computing Center of Southeast University for providing the facility support on the numerical calculations in this paper.

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

| For CTC |
| --- |
| the number of row the current cell is located in |
| the number of column the current cell is located in |
| whether the string of the current cell is unique in the current column |
| the string type of the current cell |
| whether the current cell is from the merged cell |

| For TTC |
| --- |
| the number of rows of the current table |
| the number of columns of the current table |

Figure 9: Manual features.

## A  Correspondence with Different Taxonomies and More Details

In this section, we comprehensively describe the correspondence between the various taxonomies.

The annotation taxonomy employed in *TURL* focuses on annotating the subject column within tables. For instance, in Fig. 1 (a), the subject column of the table is the second column titled "language". The correspondence between the taxonomy of *TURL* and ours is illustrated in Fig. 8. The taxonomy of this *WebSheet* is nearly identical to ours, except that they do not explicitly annotate unlabeled cells with **value**. Regarding the *WCC* dataset, we classify the tables categorized as **Matrix** into the category of **Relational** tables. This decision is made due to the absence of clear boundaries between these two types, as they share similar cell types.

To ensure data quality, we applied filters to remove certain tables. This includes tables with irregular layouts, where rows do not have the same number of cells as other rows, as well as tables with pictures, huge tables, and repeated tables. The filtering process was implemented through some rules.

|  |  | is located in header rows? | |
| --- | --- | --- | --- |
|  |  | √ | × |
| is located in the subject column? | √ | indexName | index |
|  | × | valueName | value |

Figure 8: Manual features.

## B  Manual Features

The main manual features used in our work are listed in Fig 9.

## C  Hyper-parameters

The semantic features of nodes are first output by Roberta and then reduced dimensions to 512 through PCA. We employ a 3-layer GIN architecture as the backbone whose hidden dimension is set as 1024. The activation function of GIN is ReLu. For pre-training, we set the learning rate as 0.01, the weight decay as 1e-5, and the dropout as 0.5. We set the $readout$ function as mean pooling.