# OpenReview forum: "TabPrompt: Graph-based Pre-training and Prompting for Few-shot Table Understanding"
_EMNLP/2023/Conference — EMNLP 2023 Findings_

### Official Review · Reviewer_ss3f · 2023-08-03

**Typos Grammar Style And Presentation Improvements:** It could be great to specify explictl…
**Soundness:** 3

**Excitement:**

4: Strong: This paper deepens the understanding of some phenomenon or lowers the barriers to an existing research direction.

**Paper Topic And Main Contributions:**

This paper introduces TabPrompt, a prombt based method for table understanding. It uses graph contrastive learning to generate the prompt . The graph cl model utilizes both text and topological information. Experimental results on CTC and TTC show that TabPrompt could achieve state-of-the-art results on both tasks.

**Reasons To Accept:**

1. The experimental results are fantastic, meaning the TabPrompt method is quite useful.
2. This paper opens the direction on prompts designed for tabular data.
3. The observations of tabular data on the topology are useful to future research.

**Reasons To Reject:**

1. The labeled tabular data (at least for CTC and TTC) is not under scarcity. The problem motivation is not strong enough. Maybe the authors could change the angle of this argument.
2. If I understand correctly, the baselines are used to generate prompts instead of doing the CTC/TTC tasks directly. The authors should put more direct evaluation results in Table1 and Table2 so that the comparison is fair.
3. Follow 2, if the results on CTC and TTC with prompt based learning cannot surpass existing methods, prompt based learning for tables are not well motivated.

**Reproducibility:**

3: Could reproduce the results with some difficulty. The settings of parameters are underspecified or subjectively determined; the training/evaluation data are not widely available.

**Reviewer Confidence:**

3: Pretty sure, but there's a chance I missed something. Although I have a good feel for this area in general, I did not carefully check the paper's details, e.g., the math, experimental design, or novelty.

---

> ### Author Rebuttal · Authors · 2023-08-28
>
> We are grateful to you for providing valuable feedback and suggestions. We provide explanations and clarifications for each rejection below and will revise the typos in our final version.
>
>
>
> > **R(a): The labeled tabular data (at least for CTC and TTC) is not under scarcity. The problem motivation is not strong enough. Maybe the authors could change the angle of this argument.**
>
> **A1:** Thank you for your critical consideration.
>
> While several labelled datasets are available for Table Type Classification (TTC), it is essential to note that most are not publicly accessible, as referenced in Section 4.2 of the paper [a]. Regarding Cell Type Classification (CTC), a handful of works have indeed contributed tabular annotation datasets to the public domain (see paper [b, c, d, e]). Nevertheless, these datasets cover specific domains such as general, commodity, energy, Statistical Abstract of the United States, and crime. There remains a notable absence of tabular datasets for various other domains, such as power and vehicle applications. Additionally, the variability and diversity in tabular data layouts further complicate matters. Even within the same domain, the distribution of publicly labelled datasets may not align with the specific tabular data in practical applications. Consequently, exploring few-shot scenarios is indispensable for mitigating the associated labelling costs.
>
> We will add this explanation in the final version of the paper.
>
>
>
> > **R(b): If I understand correctly, the baselines are used to generate prompts instead of doing the CTC/TTC tasks directly. The authors should put more direct evaluation results in Table1 and Table2 so that the comparison is fair.**
>
> **A2:** Thank you for your thoughtful feedback. It's possible that our explanation of how to conduct experiments on baselines in the paper wasn't sufficiently clear, leading to your misunderstanding, and we apologise for that.
>
> It is worth highlighting that all baseline approaches directly addressed the CTC/TTC tasks without engaging in prompt generation. We utilised their publicly available code as-is without modifying their inherent process. We intend to incorporate this clarification in the final version of the paper.
>
>
>
> > **R(c): Follow 2, if the results on CTC and TTC with prompt based learning cannot surpass existing methods, prompt based learning for tables are not well motivated.**
>
> **A3:** Following the previous explanation, TabPrompt outperforms all existing methods in the few-shot scenario of CTC/TTC tasks. This performance superiority underscores the motivation of tabular prompt learning.
>
>
>
> ## References
>
> [a] Wang, Zhiruo et al. “TUTA: Tree-based Transformers for Generally Structured Table Pre-training.” *Proceedings of the 27th ACM SIGKDD Conference on Knowledge Discovery & Data Mining* (2020): n. pag.
>
> [b] Deng, Xiang et al. “TURL.” *ACM SIGMOD Record* 51 (2020): 33 - 40.
>
> [c] Koci, Elvis et al. “DECO: A Dataset of Annotated Spreadsheets for Layout and Table Recognition.” *2019 International Conference on Document Analysis and Recognition (ICDAR)* (2019): 1280-1285.
>
> [d] Dong, Haoyu et al. “Semantic Structure Extraction for Spreadsheet Tables with a Multi-task Learning Architecture.” (2019).
>
> [e] Ghasemi-Gol, Majid et al. “Tabular Cell Classification Using Pre-Trained Cell Embeddings.” *2019 IEEE International Conference on Data Mining (ICDM)* (2019): 230-239.

---

### Official Review · Reviewer_MxSf · 2023-08-04

**Soundness:** 4

**Excitement:**

3: Ambivalent: It has merits (e.g., it reports state-of-the-art results, the idea is nice), but there are key weaknesses (e.g., it describes incremental work), and it can significantly benefit from another round of revision. However, I won't object to accepting it if my co-reviewers champion it.

**Missing References:**

-

**Paper Topic And Main Contributions:**

This paper proposes a new framework to capture the untapped topological in tabular data. The prompt-based learning and graph contrastive learning turns out to be effective with new SOTA results on cell type classification and table type classification.

**Questions For The Authors:**

To best use both semantic and structure information, how to unified the GNN structure with cutting-the-edge large language models?

**Reasons To Accept:**

1. It's a novel direction to leverage the topological information in tabular data.
2. Technical design is sound with prompt-based learning and graph contrastive learning.
3. Experiment results are significant
4. Good writing.

**Reasons To Reject:**

1. More tasks are desired to be evaluated., question answering such as dataset of WTQ and HiTab.


**Reproducibility:**

4: Could mostly reproduce the results, but there may be some variation because of sample variance or minor variations in their interpretation of the protocol or method.

**Reviewer Confidence:**

4: Quite sure. I tried to check the important points carefully. It's unlikely, though conceivable, that I missed something that should affect my ratings.

**Typos Grammar Style And Presentation Improvements:**

-

---

> ### Author Rebuttal · Authors · 2023-08-28
>
> We are grateful to you for providing valuable feedback and suggestions. We provide explanations and clarifications for each question below.
>
>
>
> > **R(a): More tasks are desired to be evaluated., question answering such as dataset of WTQ and HiTab.**
>
> **A1:** Thank you for your pertinent suggestion.
>
> The work centring on Tabular Structure Understanding (TU) is unsuitable for Table QA. To clarify it, we have conducted related experiments on QA tasks.
>
> Note: Our method lacks modules for the Table QA task and, therefore, cannot directly undertake this task. As a result, we integrated our method into existing strong or SOTA methods on Table QA (paper [a, b, c, d, e]) to examine whether its fusion enhances performance in the few-shot Table QA task on HiTab and WTQ datasets.
>
> Experimental results indicate only marginal improvements in the performance of the Table QA model through fusion. We speculate that the role information of table cells may not significantly enhance the model's reasoning ability under few-shot scenarios. To test this hypothesis, we conducted another experiment explicitly providing the model with the role information of each cell. Similarly, the results show that this modification yields minimal enhancements in the model's performance. Thus, we conclude that, in the context of few-shot Table QA, the model's bottleneck predominantly stems from challenges in acquiring inference-related knowledge rather than from the role information of table cells.
>
>
>
> > **Q(a): To best use both semantic and structure information, how to unified the GNN structure with cutting-the-edge large language models?**
>
> **A2**: Thank you for your thoughtful question.  As it remains an active area of exploration within the academic community, I offer some preliminary insights of my own.
>
> In the case of closed-source large models like ChatGPT, you may design prompts for the GNN's output to serve as ChatGPT's input. Alternatively, with large open-source models like LLaMA, we may treat LLaMA's outputs as embeddings containing semantic information. We then integrate these embeddings into GNN and incorporate the structural information. I've included references to two recent papers (see paper [f, g]), which provide further in-depth information.
>
>
>
> ## References
>
> [a] Cheng, Zhoujun et al. “FORTAP: Using Formulas for Numerical-Reasoning-Aware Table Pretraining.” *Annual Meeting of the Association for Computational Linguistics* (2021).
>
> [b] Zhou, Fan et al. “TaCube: Pre-computing Data Cubes for Answering Numerical-Reasoning Questions over Tabular Data.” *ArXiv* abs/2205.12682 (2022): n. pag.
>
> [c] Lin, Weizhe et al. “An Inner Table Retriever for Robust Table Question Answering.” *Annual Meeting of the Association for Computational Linguistics* (2023).
>
> [d] Yin, Pengcheng et al. “TaBERT: Pretraining for Joint Understanding of Textual and Tabular Data.” *ArXiv* abs/2005.08314 (2020): n. pag.
>
> [e] Herzig, Jonathan et al. “TaPas: Weakly Supervised Table Parsing via Pre-training.” *Annual Meeting of the Association for Computational Linguistics* (2020).
>
> [f] Yang, Lin F. et al. “ChatGPT is not Enough: Enhancing Large Language Models with Knowledge Graphs for Fact-aware Language Modeling.” *ArXiv* abs/2306.11489 (2023): n. pag.
>
> [g] Chen, Zhikai et al. “Exploring the Potential of Large Language Models (LLMs) in Learning on Graphs.” *ArXiv* abs/2307.03393 (2023): n. pag.

---

### Official Review · Reviewer_k7YW · 2023-08-05

**Typos Grammar Style And Presentation Improvements:** There are a few stylish errors like s…
**Soundness:** 3

**Excitement:**

3: Ambivalent: It has merits (e.g., it reports state-of-the-art results, the idea is nice), but there are key weaknesses (e.g., it describes incremental work), and it can significantly benefit from another round of revision. However, I won't object to accepting it if my co-reviewers champion it.

**Missing References:**

[a] Wang, Z., & Sun, J. (2022). Transtab: Learning transferable tabular transformers across tables. Advances in Neural Information Processing Systems, 35, 2902-2915.
[b] Yang, J., Gupta, A., Upadhyay, S., He, L., Goel, R., & Paul, S. (2022, May). TableFormer: Robust Transformer Modeling for Table-Text Encoding. In Proceedings of the 60th Annual Meeting of the Association for Computational Linguistics (Volume 1: Long Papers) (pp. 528-537).
[c] Ye, C., Lu, G., Wang, H., Li, L., Wu, S., Chen, G., & Zhao, J. (2023). CT-BERT: Learning Better Tabular Representations Through Cross-Table Pre-training. arXiv preprint arXiv:2307.04308.
[d] Liu, R., Yuan, S., Dai, A., Shen, L., Zhu, T., Chen, M., & He, X. (2022, October). Few-Shot Table Understanding: A Benchmark Dataset and Pre-Training Baseline. In Proceedings of the 29th International Conference on Computational Linguistics (pp. 3741-3752).

**Paper Topic And Main Contributions:**

In this paper, the authors propose to utilize prompt and graph-based methods to address the challenge of table understanding in the few-shot setting. Table values are first transformed into a tabular graph for graph network pre-training. Prompt learning can then be applied to learn few-shot tasks. Experiments are conducted on four benchmark datasets with comparisons to conventional methods. The experimental results also demonstrate that the proposed method outperforms the baseline methods in the tasks of classifying cell and table types. The main contribution of this work is mainly the novel usage of prompt learning in the table understanding problem.

**Questions For The Authors:**

Question A: The experiments are limited in terms of setup. I wonder why the authors do not conduct all two tasks on all datasets with all baseline methods.

Question B: SOTA methods are missing. It would be great to patch the performance comparisons to those methods.

Question C: Although the authors mention "significant", it would be great to conduct some significance tests like t-test, especially with high variance values.

**Reasons To Accept:**

* Levering prompt learning, especially soft prompt or prefix tuning, in table understanding is novel.
* Various benchmark datasets for the reproducibility.
* Ablation study shows the effectiveness of each component.

**Reasons To Reject:**

* Lack of comparisons to state-of-the-art methods like [a,b,c,d]
* Insufficient experiments with missing combinations of datasets, baselines, and tasks.
* References are pretty out-dated mostly with only papers until 2021.

**Reproducibility:**

4: Could mostly reproduce the results, but there may be some variation because of sample variance or minor variations in their interpretation of the protocol or method.

**Reviewer Confidence:**

4: Quite sure. I tried to check the important points carefully. It's unlikely, though conceivable, that I missed something that should affect my ratings.

---

> ### Author Rebuttal · Authors · 2023-08-28
>
> We are grateful to you for providing valuable feedback and suggestions. We provide explanations and clarifications for each question and will revise the typos in our final version.
>
>
>
> > **R(a): Lack of comparisons to state-of-the-art methods like [a,b,c,d]**
>
> **A1:** Thank you for your pertinent suggestion.
>
> The reason for not including paper[a, b, c, d] in our baselines is their incompatibility with the subtasks of Table Understanding (TU) focused by us, i.e. Table Type Classification (TTC) and Cell Type Classification (CTC).  A direct indication of this is the difference in baselines between paper[a, b, c, d] and ours. Notably, they do not encompass the SOTA *TUTA* (paper [d]) for TTC/CTC tasks.
>
> Specifically, *Transtab* (paper [a]) focuses on Machine Learning applications presented in tabular format. *TableFormer* (paper [b]) is for Table QA and Table Fact Verification. Despite *FewTPT* (paper [d]) claiming to target TU tasks, it encompasses a broader range of table-related tasks within this category, excluding TTC/CTC tasks. It's important to highlight that the "Table Classification" mentioned in *FewTPT* (paper [d]) centres on identifying header domains within tables, distinct from the focus of TTC. The differences among these tasks can be seen in the paper [f].
>
> To provide further clarification, we have performed additional experiments to elaborate:
>
> Because the methods in paper [a, b, c, d] cannot be conducted on these two tasks directly, we treat these models as encoders that output embeddings of table cells and stack layers of downstream classifiers (including an MLP layer and a pooling layer) on top of the encoders. In that case, these methods can be compared with our method.
>
> Note: *CT-BERT* (paper [c]) is contemporaneous with ours. *FewTPT* (paper [d]) is a Chinese tabular language model, thus, we pre-train it using 570k Wiki tables in English from scratch.
>
>
> |Methods|TURL (1-shot)|TURL (3-shot)|WebSheet (1-shot)|WebSheet (3-shot)|
> |:---:|:---:|:---:|:---:|:---:|
> | *Transtab* | 25.14 ± 7.33 | 29.96 ± 7.25 |14.96 ± 12.53|17.96 ± 11.78|
> | *TableFormer* | 33.68 ± 6.33 | 45.78 ± 5.88 |21.74 ± 8.95|27.75 ± 7.15|
> | *FewTPT* | 33.21± 10.22 | 44.71± 8.13 | 22.93 ± 9.75 | 28.41 ± 8.42 |
> | *TUTA* | 35.73 ± 6.81 | 47.32 ± 9.48 | 27.65 ± 13.25 | 34.77 ± 9.84 |
> | Ours | **46.82 ± 8.80** | **57.79 ± 9.01** | **30.78 ± 10.25** | **35.96 ± 9.76** |
>
> |    Methods    |   WCC (3-shot)   |   WCC (5-shot)   |
> | :-----------: | :--------------: | :--------------: |
> |  *Transtab*   |   16.71 ± 9.47   |   20.71 ± 8.81   |
> | *TableFormer* |   42.14 ± 6.73   |   50.17 ± 6.18   |
> |   *FewTPT*    |   44.97 ± 7.13   |   52.72 ± 6.42   |
> |    *TUTA*     |   46.12 ± 8.73   |   54.45 ± 8.27   |
> |     Ours      | **50.75 ± 7.18** | **57.41 ± 5.14** |
>
> Result analysis:
>
> 1. *Transtab*, *TableFormer*, and *FewTPT* exhibit inferior performance compared to *TUTA* despite some being pre-trained. This is mainly because none are designed for the TU task, leading to the absence of modules tailored to TU tasks.
> 3. Although *TableFormer*, *FewTPT*, and *TUTA* acquire table-related knowledge during their pre-training phase, the performance disparity in the downstream tasks can be attributed to the gap between the two phases. Unlike ours, these models lack an objective during the downstream task that aligns with that in the pre-training stage, resulting in their comparatively inferior performance.
>
> We will add these results to the appendix section of the paper.
>
>
>
> > **R(b): Insufficient experiments with missing combinations of datasets, baselines, and tasks.**
>
> **A2:** Thank you for pointing out this issue.
>
> While TTC and CTC fall under the broader category of TU tasks, they represent separate subtasks, leading to the distinction of datasets and baselines between them. Consequently, their corresponding datasets and baselines are not suitable for each other. (An exception to this is the TUTA method (paper [e]), which is mentioned in line 464 of our paper.) Notably, our experimental setup primarily aligns with the SOTA method, *TUTA* (paper [e]).
>
> We will explicitly add this description of the baselines in the final version of the paper.
>
>
>
> >  **R(c): References are pretty out-dated mostly with only papers until 2021.**
>
> **A3:** Thank you for your professional and rigorous feedback.
>
> The bulk of closely related work to our research emerged predominantly before 2021, with only a few instances after 2021. Despite the surge in table-related research during the recent two years (papers [a, b, c, d, f, g, h, i, j]), none of these specifically delve into exploring the CTC/TTC tasks (some are surveys, some are contemporaneous.).
>
> However, we accept the suggestion and intend to augment our final version by incorporating references to these up-to-date papers.
>
>
>
> > **Q(a): The experiments are limited in terms of setup. I wonder why the authors do not conduct all two tasks on all datasets with all baseline methods.**
>
> **A4:** The same answer as **A2** to **R(b)**.
>
>
>
> > **Q(b): SOTA methods are missing. It would be great to patch the performance comparisons to those methods.**
>
> **A5:** The same answer as **A1** to **R(a)**.
>
>
>
> > **Q(c): Although the authors mention "significant", it would be great to conduct some significance tests like t-test, especially with high variance values.**
>
> **A6:** Thank you for your insightful comment.
>
> We conducted additional experiments to support the description word "significant," which appears in the section of the Ablation study. We employ the paired Student’s t-test to discern performance changes before and after ablation. Our chosen significance level is α=0.05.
>
> |           | 3-shot CTC (TUTA) |     𝑝     | 3-shot CTC (WebSheet) |     𝑝     | 5-shot TTC (WCC) |     𝑝     |
> | :-------: | :---------------: | :-------: | :-------------------: | :-------: | :--------------: | :-------: |
> | TabPrompt |   57.79 ± 9.01    |     /     |     35.96 ± 9.76      |     /     |   57.41 ± 5.14   |     /     |
> | *w/o* cl  |   46.12 ± 10.48   | **0.027** |     31.52 ± 9.01      | **0.044** |   54.52 ± 8.76   | **0.037** |
> | *w/o* sp  |   55.74 ± 11.16   |   0.097   |     33.25 ± 9.87      |   0.314   |   56.14 ± 9.11   |   0.197   |
> | *w/o* pr  |   49.15 ± 9.38    | **0.028** |     30.18 ± 10.47     | **0.047** |   52.32 ± 13.5   | **0.049** |
>
> The provided results demonstrate that upon removing cl or pr modules, the 𝑝-values for experiments across all three datasets are consistently below the designated significance level α. This outcome is congruent with the explanations found in the paper.
>
> We will add these results to the final version of our paper.
>
>
>
> ## References
>
> [a] Wang, Z., & Sun, J. (2022). *Transtab*: Learning transferable tabular transformers across tables. Advances in Neural Information Processing Systems, 35, 2902-2915.
>
> [b] Yang, J., Gupta, A., Upadhyay, S., He, L., Goel, R., & Paul, S. (2022, May). *TableFormer*: Robust Transformer Modeling for Table-Text Encoding. In Proceedings of the 60th Annual Meeting of the Association for Computational Linguistics (Volume 1: Long Papers) (pp. 528-537).
>
> [c] Ye, C., Lu, G., Wang, H., Li, L., Wu, S., Chen, G., & Zhao, J. (2023). *CT-BERT*: Learning Better Tabular Representations Through Cross-Table Pre-training. arXiv preprint arXiv:2307.04308.
>
> [d] Liu, R., Yuan, S., Dai, A., Shen, L., Zhu, T., Chen, M., & He, X. (2022, October). Few-Shot Table Understanding: A Benchmark Dataset and Pre-Training Baseline. In Proceedings of the 29th.
>
> [e] Wang, Zhiruo et al. “*TUTA*: Tree-based Transformers for Generally Structured Table Pre-training.” *Proceedings of the 27th ACM SIGKDD Conference on Knowledge Discovery & Data Mining* (2020): n. pag.
>
> [f] Dong, Haoyu et al. “Table Pre-training: A Survey on Model Architectures, Pretraining Objectives, and Downstream Tasks.” *ArXiv* abs/2201.09745 (2022): n. pag.
>
> [g] Korini, Keti and Christian Bizer. “Column Type Annotation using ChatGPT.” *ArXiv* abs/2306.00745 (2023): n. pag.
>
> [h] Koleva, Aneta, et al. "Named Entity Recognition in Industrial Tables using Tabular Language Models." *arXiv preprint arXiv:2209.14812* (2022).
>
> [i] Badaro, Gilbert, and Paolo Papotti. "Transformers for tabular data representation: A tutorial on models and applications." *Proceedings of the VLDB Endowment* 15.12 (2022): 3746-3749.
>
> [j] Shigarov, Alexey O.. “Table understanding: Problem overview.” *Wiley Interdisciplinary Reviews: Data Mining and Knowledge Discovery* 13 (2022): n. pag.

---

### Meta-Review · Area_Chair_n9Uz · 2023-09-16

**Recommendation:** 2

**Metareview:**

This paper proposes a method for table understanding in the few-shot setting using prompt and graph-based approaches. The main contributions of the paper include the application of prompt learning, specifically soft prompt or prefix tuning, in the context of table understanding, as well as the use of graph network pre-training to transform table values into a tabular graph. The paper presents experimental results on four benchmark datasets, demonstrating the effectiveness of the proposed method in classifying cell and table types. However, there are also several weaknesses and reasons. One major weakness is the lack of comparisons to state-of-the-art methods, making it difficult to assess the performance of the proposed method relative to other approaches. Additionally, the experiments are limited in scope, with missing combinations of datasets, baselines, and tasks.

Overall, while the paper presents some interesting and novel approaches to table understanding, there are several aspects where it falls short of expectations. The authors are encouraged to address these weaknesses in their work to improve its overall impact.

---

### Decision · Program_Chairs · 2023-10-07

**Decision:**

Accept-Findings

**Comment:**

This paper proposes a method for table understanding in the few-shot setting using prompt and graph-based approaches. The main contributions of the paper include the application of prompt learning, specifically soft prompt or prefix tuning, in the context of table understanding, as well as the use of graph network pre-training to transform table values into a tabular graph. The paper presents experimental results on four benchmark datasets, demonstrating the effectiveness of the proposed method in classifying cell and table types. However, there are also several weaknesses and reasons. One major weakness is the lack of comparisons to state-of-the-art methods, making it difficult to assess the performance of the proposed method relative to other approaches. Additionally, the experiments are limited in scope, with missing combinations of datasets, baselines, and tasks.

Overall, while the paper presents some interesting and novel approaches to table understanding, there are several aspects where it falls short of expectations. The authors are encouraged to address these weaknesses in their work to improve its overall impact.